# Exploiting Pseudo-Labeling and nnU-Netv2 Inference Acceleration for Abdominal Multi-Organ and Pan-Cancer Segmentation

Ziyan Huang[1,2][0000−0002−1533−5239], Jin Ye[2][0000−0003−0667−9889], Haoyu Wang[2][0000−0002−1753−7336], Zhongying Deng[2][0000−0003−0887−7408], Tianbin Li[2][0009−0001−3617−8324], and Junjun He[2][0000−0002−1813−1784]

[1] Shanghai Jiao Tong University, Shanghai, China
ziyanhuang@sjtu.edu.cn
[2] Shanghai AI Laboratory, Shanghai, China
hejunjun@pjlab.org.cn

**Abstract.** Deep-learning based models offer powerful tools for the automatic segmentation of abdominal organs and tumors in CT scans, yet they face challenges such as limited datasets and high computational costs. The FLARE23 challenge addresses these by providing a large-scale dataset featuring both partially and fully annotated data, and by prioritizing both segmentation accuracy and computational efficiency. In this study, we adapt the winning FLARE22 strategy to FLARE23 by utilizing a two-step pseudo-labeling approach. Initially, a large model trained on datasets with complete organ annotations generates pseudo-labels for datasets that originally contain only tumor annotations. These labels are then integrated to create a comprehensive training dataset. A smaller, more efficient model is subsequently trained on this enriched dataset for deployment, targeting both tumors and organs. Our approach, utilizing the FLARE23 dataset, has achieved notable results. On the online validation leaderboard, it reached an average DSC of 89.63% for organs and 46.07% for lesions, with an average processing time of 16.1 seconds for 20 selected validation cases. In the final testing set, our model demonstrated improved performance, achieving an organ DSC of 89.98% and lesion DSC of 62.61%, while reducing the average processing time to 12.02 seconds. The code and model are publicly available at https://github.com/Ziyan-Huang/FLARE23.

**Keywords:** Medical Image Segmentation · Computational Efficiency · Abdominal Tumors

## 1 Introduction

The abdomen is a prevalent site for tumor growth. Accurate annotation of tumors and relevant abdominal organs in CT scans is essential for the diagnosis and treatment of abdominal tumors. While deep-learning-based methods ease

the task of manual annotation for radiologists, several challenges hinder their effectiveness. Firstly, there's a lack of comprehensive datasets that include annotations for both tumors and various abdominal organs. Many existing datasets focus either on organ-specific or tumor-specific annotations. Therefore, learning accurate segmentations from these partially labeled and unlabeled datasets remains a challenge. Second, while state-of-the-art solutions like nnU-Net offer robust performance, they are often computationally intensive, thereby limiting their clinical utility. Recognizing these challenges, the FLARE23 challenge has been established. It offers a large-scale dataset that includes both partially annotated and unlabeled data, and it focuses on both segmentation accuracy and efficiency as evaluation metrics.

Given the challenge of insufficiently fully annotated datasets, semi-supervised and partial-label methods have increasingly garnered attention in the field of medical image segmentation. DoDNet [22] employs a dynamic on-demand network with a shared encoder-decoder architecture and a unique segmentation head, efficiently segmenting multiple organs and tumors from partially labeled datasets. In a similar vein, the Universal Model [10] employs Contrastive Language–Image Pretraining (CLIP) [17] to extract semantic relationships between abdominal structures, achieving high performance across multiple datasets. MultiTalent [19] adopts a multi-dataset learning approach, incorporating a class and dataset adaptive loss function to handle varying dataset characteristics and overlapping classes. As for using unlabeled data, the FLARE22 championship solution [8] demonstrates significant performance gains through pseudo-labeling and label-filtering techniques on unlabeled data. It also introduces a highly efficient, optimized version of nnU-Net [9]. However, the advent of nnU-Net v2, which excels in code usability, calls for new acceleration techniques tailored to this updated framework.

In this study, we extend the winning strategy of FLARE22 for application in the FLARE23 challenge by leveraging pseudo-labeling techniques. We employ partially-annotated and unannotated data to create datasets with comprehensive pseudo-labels. For efficiency, two different model sizes are utilized: a larger model for generating pseudo-labels and a smaller, deployable model for the final application. Specifically, we categorize the partially-labeled data into two main groups: one with comprehensive annotations for 13 types of abdominal organs, and another focused on tumor annotations. The pseudo-labeling process is executed in two stages. Initially, a larger model is trained on data with complete organ annotations to specialize in segmenting the 13 abdominal organs. This model then pseudo-labels organ annotations for datasets initially containing only tumor annotations. Subsequently, a full-annotation dataset is created by combining the new organ annotations with existing tumor annotations. A smaller, more efficient model is then trained on this comprehensive dataset for the final deployment. In this manner, we successfully generate organ and tumor labels for all 4000 complete datasets, while optimizing the inference speed of the latest nnU-Netv2 framework.

## 2   Method

### 2.1   Preprocessing

We employ the nnU-Net framework's default preprocessing. For anisotropic data resampling, trilinear interpolation is used in the axial plane and linear interpolation in the sagittal direction. Intensity normalization is performed by clipping values to the 0.5% (-970.0) and 99.5% (279.0) Hounsfield Unit levels, followed by z-normalization using a mean of 80.3 and a standard deviation of 141.4.

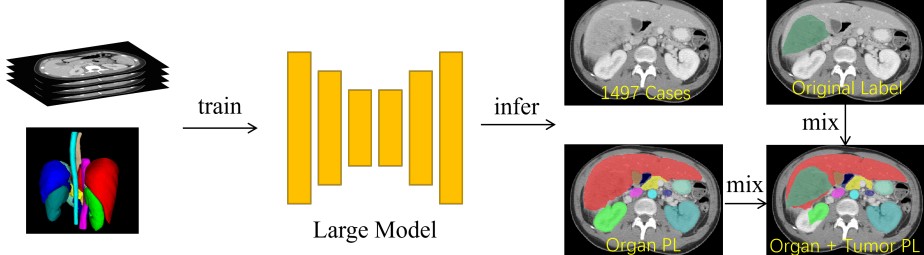

(1) Phase 1: Pseudo-Labeling Organs in 1497 Tumor-Annotated Images

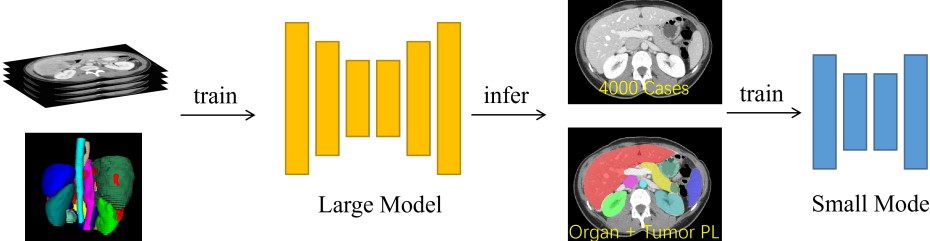

(2) Phase 2: Pseudo-Labeling Organs and Tumor in All 4000 Images

**Fig. 1.** Pipeline of our two-stage pseudo-labeling method. In the first stage, a large model trained for segmenting 13 organs assigns pseudo-labels to 1,497 tumor-annotated images. These images then receive combined organ and tumor labels. In the second stage, another large model trained on these 1,497 images assigns pseudo-labels for the remaining dataset. Finally, a small model is trained using the complete 4,000-image dataset.

### 2.2   Proposed Method

Inspired by the winning solution of FLARE 2022 from Huang et al. [8], we implement a two-stage approach for generating pseudo-labels and eventual model deployment. We employ varying sizes of STU-Net architectures [7] for these stages. For a comprehensive overview of our method, please refer to Figure 1.

**STU-Net with different scales** Figure 2 illustrates the architecture of our STU-Net, which serves as an extendable and transferable version of the nnU-Net. We achieve this by fixing certain configurations within the nnU-Net framework, adding residual connections to the basic blocks, and modifying the up-sampling and down-sampling techniques. In our experiments, we employed STU-Net-L for the generation of pseudo-labels and utilized STU-Net-B for the final inference deployment. These specific configurations are elaborated in the Table 1.

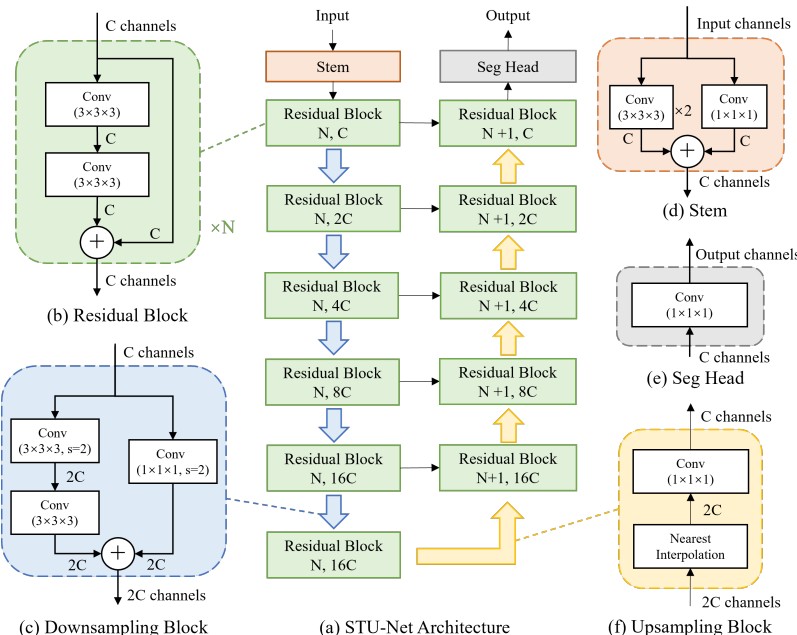

**Fig. 2.** Illustration of our STU-Net architecture which is built upon the nnU-Net architecture with several modifications to enhance its scalability and transferability. (a) An overview of the STU-Net architecture. The blue arrows denote downsampling while the yellow ones represent upsampling. (b) Residual blocks to achieve a large-scale model. (c) Downsampling in the first residual block of each encoder stage. (d-e) Stem and segmentation head for channel conversion of input and output. (f) Weight-free interpolation for upsampling, which effectively addresses the issue of weight mismatch across different tasks.

Loss function: we use the summation between Dice loss and cross-entropy loss because compound loss functions have been proven to be robust in various medical image segmentation tasks [11].

**Handling Partially-Labeled and Unlabeled Data** We divide the 2,200 partially-labeled FLARE23 images into three main categories, as summarized in

**Table 1.** Configurations of STU-Net-L and STU-Net-B models. Depth indicates the number of residual blocks at each resolution stage, and width denotes the channel count at each stage.

| Model | depth | width | Params (M) | FLOPs (T) |
|---|---|---|---|---|
| STU-Net-B | (1,1,1,1,1,1) | (32,64,128,256,512,512) | 58.26 | 0.51 |
| STU-Net-L | (2,2,2,2,2,2) | (64,128,256,512,1024,1024) | 440.30 | 3.81 |

**Table 2.** Categorization of Partially-labeled Data in FLARE23 Dataset: 2,200 images grouped into three categories

| Category | Number of Cases |
|---|---|
| 13-organs, no tumor | 250 |
| Tumor, some 5-organs | 1,497 |
| Only 5-organs | 453 |

Table 2. We particularly focus on the subsets containing 250 and 1,497 images. Initially, a large STU-Net model (STU-Net-L) is trained on the 250 images annotated for 13 abdominal organs. This model is then applied to the set of 1,497 images, augmenting the organ annotations while preserving existing tumor labels.

For consistency, all pseudo-labels are generated by a large STU-Net model (STU-Net-L). Using the augmented 1,497-image set from the first stage, we train another STU-Net-L model to generate pseudo-labels for the remaining dataset. In the event of annotation conflicts, the original labels are preserved. Ultimately, we employ the fully augmented 4,000-image dataset to train a smaller STU-Net model (STU-Net-B) for efficient deployment and inference.

**Inference Accelaration Based on nnU-Netv2** We build our efficient inference code upon the popular nnU-Net framework, particularly its latest version, v2. Several optimizations are made to accelerate the inference process. These include using larger target spacing, eliminating the cropping stage, and replacing the resampling function in skimage with torch.nn.interpolate to reduce computational load. Given that the FLARE2023 competition performs inference on a per-image basis, we transition from multi-threading to single-threaded inference to better align with the competition's structure. Additionally, we adopt last year's championship-winning efficient inference strategy, which involves skipping certain patches during patch-based inference.

### 2.3 Post-processing

During the pseudo-labeling generation phase, we employed Testing Time Augmentation (TTA) along the anatomical axes: sagittal, coronal, and axial, to enhance the quality of the generated labels.

However, in the final submission, we skipped post-processing for computational efficiency. The model's raw outputs serve as the final segmentation results without further modification.

## 3    Experiments

### 3.1    Dataset and evaluation measures

The FLARE 2023 challenge is an extension of the FLARE 2021-2022 [13][14], aiming to promote the development of foundation models in abdominal disease analysis. The segmentation targets cover 13 organs and various abdominal lesions. The training dataset is curated from more than 30 medical centers under the license permission, including TCIA [2], LiTS [1], MSD [18], KiTS [5,6], autoPET [4,3], TotalSegmentator [20], and AbdomenCT-1K [15]. The training set includes 4000 abdomen CT scans where 2200 CT scans with partial labels and 1800 CT scans without labels. The validation and testing sets include 100 and 400 CT scans, respectively, which cover various abdominal cancer types, such as liver cancer, kidney cancer, pancreas cancer, colon cancer, gastric cancer, and so on. The organ annotation process used ITK-SNAP [21], nnU-Net [9], and MedSAM [12].

The evaluation metrics encompass two accuracy measures—Dice Similarity Coefficient (DSC) and Normalized Surface Dice (NSD)—alongside two efficiency measures—running time and area under the GPU memory-time curve. These metrics collectively contribute to the ranking computation. Furthermore, the running time and GPU memory consumption are considered within tolerances of 15 seconds and 4 GB, respectively.

### 3.2    Implementation details

**Environment settings** The development environments and requirements are presented in Table 3.

**Training protocols** To handle partially labeled and unlabeled data, we utilize the preprocessing and pseudo-labeling scheme discussed earlier. Alongside, we adopt extensive data augmentation techniques, including rotations, elastic deformations, and random cropping, to enhance our models' generalization capabilities. For training, a patch-based approach is employed. We use a balanced sampling mechanism in our patch sampling strategy to ensure equal representation of each class in each batch, effectively countering class imbalance issues. We do not conduct model selection.

**Table 3.** Development environments and requirements.

| | |
|---|---|
| System | CentOS 7 |
| CPU | Intel(R) Xeon(R) Platinum 8369B CPU @ 2.90GHz |
| RAM | 32×4GB; 2.67MT/s |
| GPU (number and type) | one NVIDIA A100 80G |
| CUDA version | 11.7 |
| Programming language | Python 3.9 |
| Deep learning framework | torch 2.0 |
| Specific dependencies | nnU-Net 2.1 |
| Code | https://github.com/Ziyan-Huang/FLARE23 |

**Table 4.** Training protocols for the STU-Net-L model.

| | |
|---|---|
| Network initialization | He |
| Batch size | 2 |
| Patch size | 48×192×192 |
| Total epochs | 2000 |
| Optimizer | SGD with nesterov momentum ($\mu = 0.99$) |
| Initial learning rate (lr) | 0.01 |
| Lr decay schedule | poly decay |
| Training time | 48 hours |
| Loss function | Dice Loss + Cross Entropy |
| Number of model parameters | 440M[3] |
| Number of flops | 3.81T[4] |
| $CO_2$eq | 114.02 Kg[5] |

**Table 5.** Training protocols for the STU-Net-B model.

| | |
|---|---|
| Network initialization | He |
| Batch size | 2 |
| Patch size | 48×128×160 |
| Total epochs | 2000 |
| Optimizer | SGD with nesterov momentum ($\mu = 0.99$) |
| Initial learning rate (lr) | 0.01 |
| Lr decay schedule | poly decay |
| Training time | 24 hours |
| Loss function | Dice Loss + Cross Entropy |
| Number of model parameters | 58M[6] |
| Number of flops | 510G[7] |
| $CO_2$eq | 17.08 Kg[8] |

**Table 6.** Quantitative evaluation results.

| Target | Public Validation | | Online Validation | | Testing | |
|---|---|---|---|---|---|---|
| | DSC(%) | NSD(%) | DSC(%) | NSD(%) | DSC(%) | NSD (%) |
| Liver | 97.70 ± 0.51 | 99.37 ± 0.48 | 97.61 | 99.29 | 96.57 | 98.20 |
| Right Kidney | 94.96 ± 5.19 | 96.84 ± 6.50 | 93.78 | 95.96 | 93.91 | 95.22 |
| Spleen | 96.64 ± 0.85 | 99.20 ± 1.32 | 96.67 | 99.41 | 96.09 | 98.47 |
| Pancreas | 87.07 ± 4.85 | 97.71 ± 2.95 | 85.82 | 96.97 | 90.37 | 98.20 |
| Aorta | 94.17 ± 2.18 | 98.64 ± 2.67 | 94.33 | 98.74 | 94.62 | 99.49 |
| Inferior vena cava | 92.84 ± 2.34 | 97.47 ± 2.32 | 92.81 | 97.34 | 93.34 | 98.40 |
| Right adrenal gland | 79.18 ± 12.52 | 94.93 ± 13.80 | 79.99 | 95.80 | 79.17 | 95.33 |
| Left adrenal gland | 80.41 ± 6.70 | 95.70 ± 4.18 | 79.94 | 94.97 | 80.00 | 95.16 |
| Gallbladder | 85.91 ± 19.62 | 88.06 ± 20.92 | 88.27 | 89.93 | 84.12 | 87.67 |
| Esophagus | 82.04 ± 15.17 | 93.95 ± 14.49 | 82.81 | 94.93 | 88.21 | 98.95 |
| Stomach | 93.92 ± 2.91 | 98.24 ± 3.25 | 94.19 | 98.34 | 93.53 | 98.09 |
| Duodenum | 84.65 ± 6.22 | 96.21 ± 4.65 | 85.47 | 96.75 | 88.37 | 98.01 |
| Left kidney | 94.00 ± 6.88 | 95.41 ± 9.33 | 93.46 | 95.59 | 92.96 | 94.62 |
| Tumor | 53.35 ± 34.22 | 45.24 ± 30.74 | 46.07 | 39.17 | 62.61 | 52.15 |
| Average | 86.92 ± 8.58 | 92.64 ± 8.40 | 86.52 | 92.37 | 88.13 | 93.43 |

**Table 7.** Performance Comparison: Partially Labeled vs. Total Data

| Training Data | Organ DSC | Organ NSD | Tumor DSC | Tumor NSD |
|---|---|---|---|---|
| 2200 Partial Label | 89.45 | 96.20 | 45.91 | 40.04 |
| 4000 Total | 89.63 | 96.46 | 46.07 | 39.17 |

## 4   Results and discussion

### 4.1   Quantitative results on validation set

Our final model's performance metrics are summarized in Table 6. Due to limitations in the online submission system, we present the average results obtained solely on a publicly labeled validation set of 50 cases.

Additionally, we conducted an ablation study to assess the impact of utilizing unlabeled data. Specifically, we compared the performance of STU-Net-L models trained on two different datasets: one with 2,200 partially labeled images and another with a total of 4,000 images. The results from the online leaderboard for both training scenarios are detailed in Table 7. As indicated by the data in Table 7, the inclusion of an extra 1,800 unlabeled images led to only minimal changes in performance metrics.

### 4.2   Qualitative results on validation set

Qualitative results of two examples with good segmentation results and two examples with bad segmentation results in the validation set are shown in Figure 3. As can be seen from the figure, our model performs well in segmenting larger tumors that are situated on organs. However, for smaller tumors that are not

**Table 8.** Quantitative evaluation of segmentation efficiency in terms of the running them and GPU memory consumption. Total GPU denotes the area under GPU Memory-Time curve. Evaluation GPU platform: NVIDIA QUADRO RTX5000 (16G).

| Case ID | Image Size | Running Time (s) | Max GPU (MB) | Total GPU (MB) |
|---|---|---|---|---|
| 0001 | (512, 512, 55) | 22.03 | 2836 | 14975 |
| 0051 | (512, 512, 100) | 13.02 | 3144 | 16366 |
| 0017 | (512, 512, 150) | 28.82 | 3212 | 23825 |
| 0019 | (512, 512, 215) | 20.33 | 2974 | 16467 |
| 0099 | (512, 512, 334) | 14.13 | 3140 | 16904 |
| 0063 | (512, 512, 448) | 16.51 | 3210 | 19762 |
| 0048 | (512, 512, 499) | 16.17 | 3180 | 17090 |
| 0029 | (512, 512, 554) | 19.85 | 3394 | 23710 |

located on organs, the model tends to miss the segmentation. Further investigation reveals that the model's limitations on smaller, isolated tumors could be attributed to the initial training set, which mainly consists of larger, organ-associated tumors.

### 4.3 Segmentation Efficiency Results on Validation Set

Efficiency results for multiple validation cases are presented in Table 8. As observed, our algorithm completes the segmentation in less than 30 seconds for all cases, with the majority finishing within 20 seconds. Additionally, the GPU memory consumption stays below 4GB. These results demonstrate that our model not only performs well in terms of accuracy but also excels in computational efficiency.

### 4.4 Segmentation Efficiency Ablation

We conduct our experiments on a consistent setup featuring an Intel Core i9-13900K CPU and an NVIDIA RTX 4090 GPU. We analyze the time efficiency for Case FLARE23Ts_0063, a typically time-consuming case, with dimensions $448 \times 512 \times 512$ and spacing $1.5 \times 0.875 \times 0.875$.

Figure 4 illustrates the time consumption for various segmentation phases both before and after optimization. Before optimization, the process was most time-consuming in "Resample Logits," taking up to 54 seconds. After applying our optimization techniques, the time spent on this phase dramatically dropped to just 0.06 seconds. Similarly, "Sliding Window Inference" was reduced from 13.4 to 2 seconds.

Overall, the total time was reduced from approximately 92 seconds to about 11 seconds, demonstrating an 8-fold efficiency improvement in the segmentation process.

### 4.5 Results on final testing set

We represent our final testing set in Table 9.

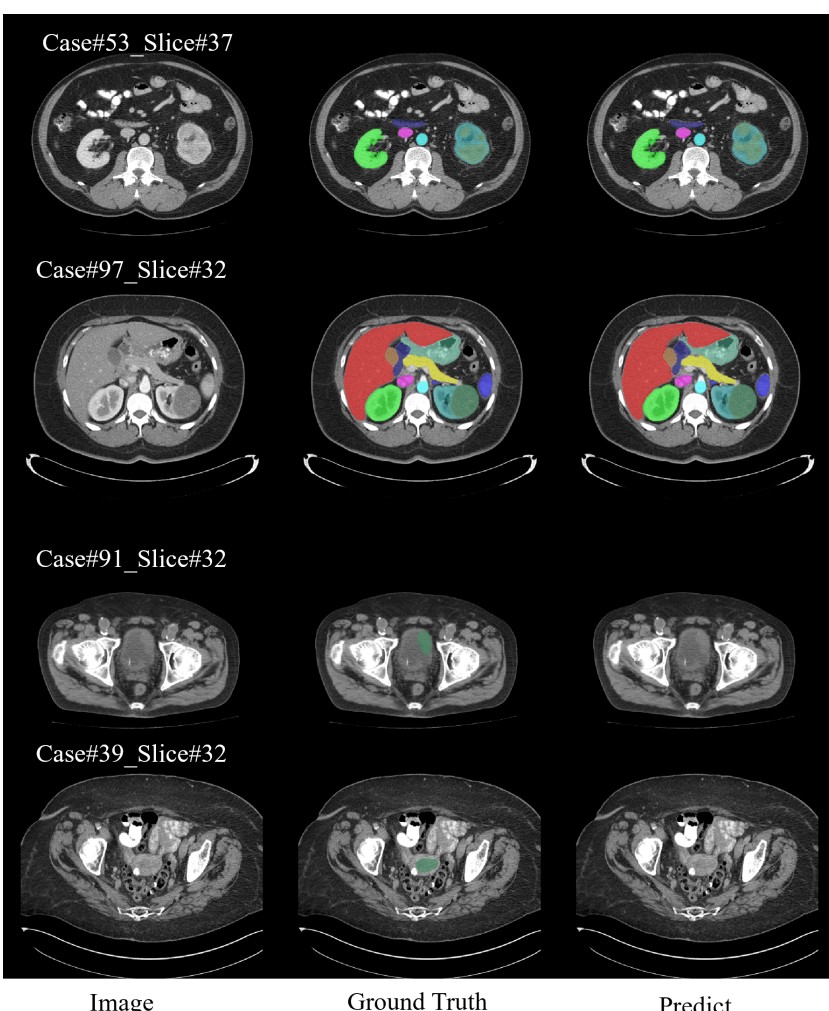

**Fig. 3.** Qualitative results of two examples with good segmentation results and two examples with bad segmentation results in the validation set.

| Organ DSC | Organ NSD | Lesion DSC | Lesion NSD | Time | GPU Memory |
|-----------|-----------|------------|------------|------|------------|
| 89.98 | 96.53 | 62.61 | 52.15 | 12.02 | 12033 |

**Table 9.** Results on final testing set

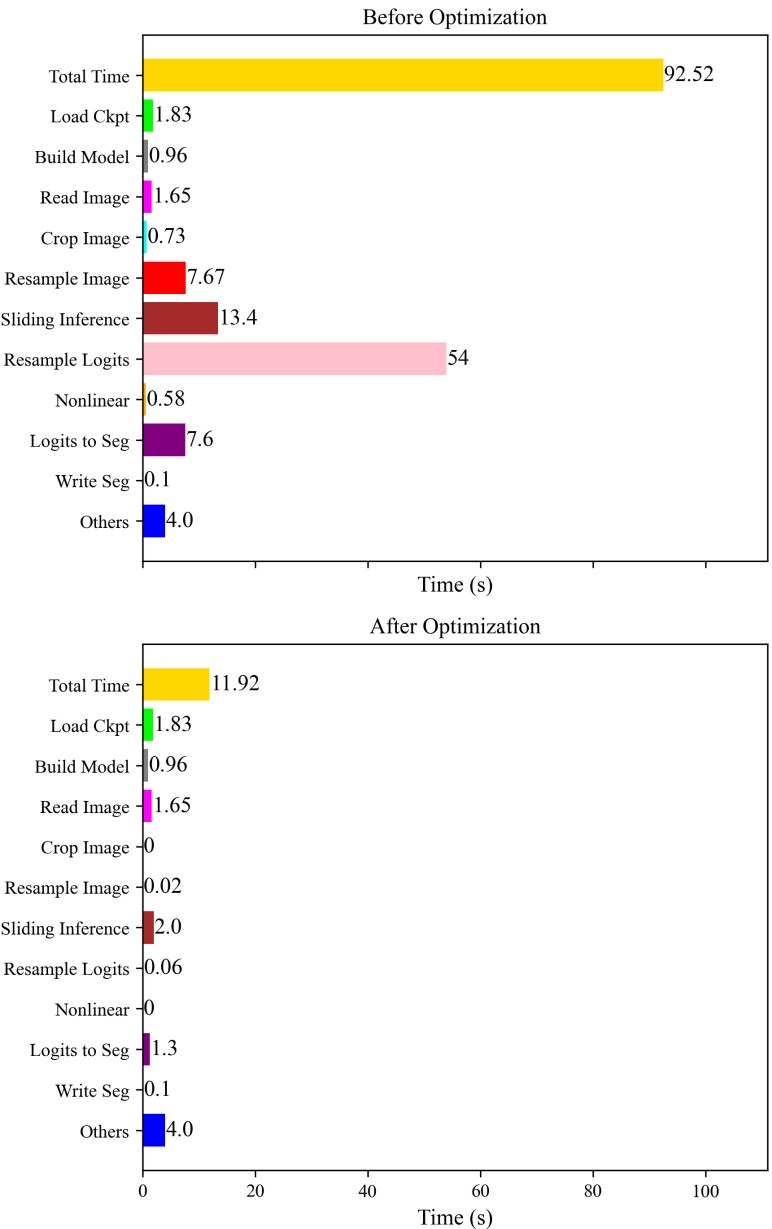

**Fig. 4.** Comparison of time consumption for various segmentation phases before and after optimization. The case analyzed is FLARE23Ts_0063, a typically time-consuming case.

### 4.6   Limitation and Future Work

One of the limitations of our approach lies in the segmentation of tumors, where a notable number of false negatives and missed detections have been observed. This issue is partly attributed to our data processing methodology, where cases marked with tumors were not comprehensively annotated. We operated under the assumption that all tumors were identified in such cases, which was a misstep. A more meticulous approach to tumor annotation is essential to overcome this challenge. Additionally, in our pursuit of accelerating the process, we opted to resize the segmentation results instead of the logits. This decision led to a significant decline in accuracy. Future work will focus on augmenting the training data to include more varied tumor types and sizes for improved generalization, alongside refining our data processing and segmentation methods to enhance precision and reliability.

## 5   Conclusion

The primary focus of our study has been to address the issue of partially labeled data in abdominal multi-organ and tumor segmentation. We explored a pseudo-labeling strategy to efficiently handle this challenge, breaking it down into a two-step process focused on separate organ and tumor annotations. Additionally, to reconcile the trade-off between accuracy and computational efficiency, we optimized the nnU-Netv2 segmentation framework. As a result, we have developed a methodology that is both accurate and efficient.

**Acknowledgements** The authors of this paper declare that the segmentation method they implemented for participation in the FLARE 2023 challenge has not used any pre-trained models nor additional datasets other than those provided by the organizers. The proposed solution is fully automatic without any manual intervention. We thank all the data owners for making the CT scans publicly available and CodaLab [16] for hosting the challenge platform.

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

**Table 10.** Checklist Table. Please fill out this checklist table in the answer column.

| Requirements | Answer |
|---|---|
| A meaningful title | Yes |
| The number of authors ($\leq 6$) | 6 |
| Author affiliations and ORCID | Yes |
| Corresponding author email is presented | Yes |
| Validation scores are presented in the abstract | Yes |
| Introduction includes at least three parts: background, related work, and motivation | Yes |
| A pipeline/network figure is provided | 3 |
| Pre-processing | 3 |
| Strategies to use the partial label | 5 |
| Strategies to use the unlabeled images. | 5 |
| Strategies to improve model inference | 5 |
| Post-processing | 5 |
| Dataset and evaluation metric section is presented | 6 |
| Environment setting table is provided | 7 |
| Training protocol table is provided | 7 |
| Ablation study | 8,11 |
| Efficiency evaluation results are provided | 9 |
| Visualized segmentation example is provided | 9 |
| Limitation and future work are presented | Yes |
| Reference format is consistent. | Yes |