# OpenReview forum: "Exploiting Pseudo-Labeling and nnU-Netv2 Inference Acceleration for Abdominal Multi-Organ and Pan-Cancer Segmentation"
_MICCAI.org/2023/FLARE — Submitted to FLARE 2023_

### Official Review · Reviewer_2hGk · 2023-10-03
**Review for“Exploiting Pseudo-Labeling and nnU-Netv2 Inference Acceleration for Abdominal Multi-Organ and Pan-Cancer Segmentation“**

**Rating:** 9
**Confidence:** 4

**Review:**

Pros:
1.The article has a complete structure.
2.This method trains large models on datasets with complete organ annotations to generate pseudo-labels for datasets that initially only contain tumor annotations. These labels are then integrated to create a comprehensive training dataset. Subsequently, a smaller, more efficient model is trained on this enriched dataset for deployment, targeting both tumors and organs. This approach achieves an average DSC of 89.63% in multi-organ segmentation and an average DSC of 46.07% in pan-tumor segmentation. The average inference time per case on the validation set is 16.1 seconds.

Cons: Fig.2 do not use the Times New Roman font.

---

### Official Review · Reviewer_jzJf · 2023-10-03
**Review for “Exploiting Pseudo-Labeling and nnU-Netv2 Inference Acceleration for Abdominal Multi-Organ and Pan-Cancer Segmentation“**

**Rating:** 9
**Confidence:** 4

**Review:**

The paper is a nice work, which is well-organized and proposes a two-stage backbone to achieve efficient and effective segmentation of oragns and tumors.

However, Fig.3 didn't show the results of ablation research.

---

> ### Comment · Reviewer_jzJf · 2023-11-30
> **2nd round Review**
>
> This article is generally very good, except for Figure 3 which does not provide the results of the ablation research.
>
> In addition, this article has made a significant contribution to abdominal organ segmentation and the open source community by publicly disclosing their code and model, and the effectiveness of the model is very good. It is recommended to accept it.

---

### Official Review · Reviewer_fZDb · 2023-10-04
**This study adapts the winning FLARE22 strategy to FLARE23 by utilizing a two-step pseudo-labelling approach and achieves an average DSC score of 89.63% for organs and 46.07% for tumours on the online validation leaderboard.**

**Rating:** 8
**Confidence:** 5

**Review:**

The method proposed in this study is an extension of the FLAR22 winning method, achieving advanced performance and efficiency in both organ and tumour segmentation tasks. This article is well written, showing the details of the method, while fully complying with the format requirements submitted. This paper also releases the code and models used, contributing to community research.

---

### Official Review · Reviewer_vhTe · 2023-10-04
**In this paper, the authors adapt the winning FLARE22 strategy to FLARE23 by utilizing a two-step pseudo-labelling approach and achieves an average DSC score of 89.63% for organs and 46.07% for tumours on the online validation leaderboard.**

**Rating:** 8
**Confidence:** 5

**Review:**

The method proposed in this study is an extension of the FLAR22 winning method and has achieved advanced performance and efficiency in both organ and tumour segmentation tasks. This article is well written, showing the details of the method, while fully complying with the submitted format requirements. This article also releases the code and models used, contributing to community research.

---

### Official Review · Reviewer_Zds7 · 2023-10-16
**Review for “Exploiting Pseudo-Labeling and nnU-Netv2 Inference Acceleration for Abdominal Multi-Organ and Pan-Cancer Segmentation“**

**Rating:** 9
**Confidence:** 5

**Review:**

The authors propose an efficient and effective approach for segmenting abdominal organs and tumors in CT scans. They adapt the winning FLARE22 strategy to FLARE23 by utilizing a two-step pseudo-labeling approach. The results demonstrate high accuracy and computational efficiency. The code and model are publicly available. Overall, this work is valuable for the medical imaging community.

---

### Official Review · Reviewer_3LsU · 2023-10-23
**Review for “Exploiting Pseudo-Labeling and nnU-Netv2 Inference Acceleration for Abdominal Multi-Organ and Pan-Cancer Segmentation“**

**Rating:** 8
**Confidence:** 5

**Review:**

This paper presented a two-step pseudo-labeling technique for dataset enrichment and achieved an average DSC score of 89.63% for organs and 46.07% for tumors on the online validation leaderboard. The provision of the code and model on GitHub enhances the transparency and reproducibility of the research. I, therefore, would recommend accepting this paper.

---

### Comment · Reviewer_fZDb · 2023-11-30

I maintain the same comment as before:

---"The method proposed in this study is an extension of the FLAR22 winning method, achieving advanced performance and efficiency in both organ and tumour segmentation tasks. This article is well written, showing the details of the method, while fully complying with the format requirements submitted. This paper also releases the code and models used, contributing to community research."

---

### Decision · Program_Chairs · 2023-10-24

Accept